# Immunoinformatic Study of Recombinant LigA/BCon1-5 Antigen and Evaluation of Its Diagnostic Potential in Primary and Secondary Binding Tests for Serodiagnosis of Porcine Leptospirosis

**DOI:** 10.3390/pathogens10091082

**Published:** 2021-08-26

**Authors:** Sujit Kumar Behera, Thankappan Sabarinath, Prasanta Kumar K. Mishra, Yosef Deneke, Ashok Kumar, Shanmugam ChandraSekar, Kuppusamy Senthilkumar, MedRam Verma, Balasubramanian Ganesh, Amol Gurav, Abhishek Hota

**Affiliations:** 1Department of Epidemiology & Public Health, Central University of Tamil Nadu, Tiruvarur 610001, India; sujitkumar@cutn.ac.in; 2Clinical Bacteriological Laboratory, Indian Council of Agricultural Research—Indian Veterinary Research Institute, Mukteshwar, Nainital 263138, India; 3Faculty of Veterinary and Animal Sciences, Rajiv Gandhi South Campus, Banaras Hindu University, Mirzapur 231001, India; pkkm@bhu.ac.in; 4School of Veterinary Medicine, Jimma University, Jimma 378, Ethiopia; yosef.deneke@ju.edu.et; 5Krishi Bhawan, Indian Council of Agricultural Research, New Delhi 110001, India; ashok.kumar21@icar.gov.in; 6Biochemistry Laboratory, Indian Council of Agricultural Research—Indian Veterinary Research Institute, Mukteshwar, Nainital 263138, India; s.sekar@icar.gov.in; 7Zoonoses Research Laboratory, Tamil Nadu Veterinary and Animal Sciences University, Chennai 600051, India; senthilkumar.k@tanuvas.ac.in; 8Livestock Economics & Statistics Division, Indian Council of Agricultural Research—Indian Veterinary Research Institute, Bareilly 243122, India; mr.verma@icar.gov.in; 9ICMR-National Institute of Epidemiology, Ayapakkam, Chennai 600077, India; ganesh@nie.gov.in; 10Temperate Animal Husbandry Division, ICAR—Indian Veterinary Research Institute (IVRI), Mukteshwar, Nainital 263138, India; amol.gurav@icar.gov.in; 11Department of Animal Science, Centurion University of Technology and Management, Paralakhemundi 761211, India; abhishek.hota@cutm.ac.in

**Keywords:** bioinformatics tools, Dot ELISA dipstick test, in silico prediction, latex agglutination test, leptospirosis, microscopic agglutination test

## Abstract

Leptospirosis is responsible for hampering the productivity of swine husbandry worldwide. The aim of this study was to assess the efficacy of bioinformatics tools in predicting the three-dimensional structure and immunogenicity of recombinant LigBCon1-5 (rLigBCon1-5) antigen. A battery of bioinformatics tools such as I-TASSER, ProSA and SAVES v6.0 were used for the prediction and assessment of the predicted structure of rLigBCon1-5 antigen. Bepipred-2.0, DiscoTope v2.0 and ElliPro servers were used to predict linear and conformational epitopes while T-cell epitopes were predicted using NetMHCpan 4.1 and IEDB recommended 2.22 method for MHC Class I and II peptides respectively. The results obtained using various in silico methods were then compared with wet lab experiments comprising of both primary (IgG Dot ELISA Dipstick test) and secondary-binding assays (Latex Agglutination Test [LAT]) to screen 1153 porcine serum samples. The three-dimensional structure of rLigA/BCon1-5 protein as predicted by I-TASSER was found to be reliable by Ramachandran Plot and ProSA. The ElliPro server suggested 10 and three potential linear and conformational B-cell-epitopes, respectively, on the peptide backbone of the rLigA/BCon1-5 protein. The DiscoTope prediction server suggested 47 amino acid residues to be part of B-cell antigen. Ten of the most efficient peptides for MHC-I and II grooves were predicted by NetMHCpan 4.1 and IEDB recommended 2.22 method, respectively. Of these, three peptides can serve dual functions as it can fit both MHC I and II grooves, thereby eliciting both humoral-and cell-mediated immune responses. The prediction of these computational approaches proved to be reliable since rLigBCon1-5 antigen-based IgG Dot ELISA Dipstick test and LAT gave results in concordance to gold standard test, the Microscopic Agglutination Test (MAT), for serodiagnosis of leptospirosis. Both the IgG Dot ELISA Dipstick test and LAT were serodiagnostic assays ideally suited for peripheral level of animal health care system as “point of care” tests for the detection of porcine leptospirosis.

## 1. Introduction

Leptospirosis causes huge economic burden to swine husbandry, primarily due to reproductive losses ranging from abortion, decreased number of piglets per litter, birth of runt piglets, increased weaning-to-oestrus interval, agalactia and so-called SMEDI syndrome (Stillbirth; Mummification/Maceration; Embryo Death; Infertility) [1,2]. Swine act as maintenance hosts for leptospiral serovars such as Bratislava, Pomona and Tarassovi Mitis and Muenchen [3,4], while serovars such as Canicola, Grippotyphosa, Hardjo and Icterohaemorrhagiae commonly cause incidental infections in pigs [5,6].

The gold standard serological test for detecting leptospirosis in humans and animals is the microscopic agglutination test (MAT) [7]. The advantage of MAT is that it provides information about *Leptospira* serogroups circulating in various animal species in a geographical area [8]. However, the inherent flaws of MAT have forced researchers to search for alternative field-oriented tests [9].

A noteworthy example for field-oriented spot test is Latex Agglutination Test (LAT) which is a highly economical, rapid screening penside test ideally suited for large-scale screening of sera samples in endemic areas without using any sophisticated equipment [10]. Another field-oriented test is Dot-ELISA which is a simple, inexpensive and user-friendly test that can be used either in a single test format or for screening a large number of sera specimens [11,12].

The advent of recombinant DNA technology has enabled the use of outer membrane proteins (OMPs) such as LipL32 [13], LipL21 [14], OmpL1 and LipL41 [15] and Loa22 [16], which are found ubiquitously in pathogenic leptospires to be used in molecular diagnostic assays. The recombinant OMPs have the ability to overcome various shortcomings of whole leptospira antigen-based assays such as MAT when employed in molecular diagnostic assays due to the fact that recombinant proteins lack nonspecific moieties found in whole-cell preparations [17,18].

Pathogenic *Leptospira* species express surface-exposed leptospiral Immunoglobulin like proteins(LigA and LigB) possessing bacterial immunoglobulin-like (Big) domains, which play a crucial role in host cell attachment and invasion during leptospiral pathogenesis [19].Clustered and regularly interspaced short palindromic repeat interference CRISPRi)-mediated LigA and LigB silencing in pathogenic *L. interrogans* drastically reduced bacterial survival upon exposure to bovine serum, confirming the role of these proteins in resistance to serum, and thus account for the virulence attenuation that resulted in asymptomatic infection in a hamster model when Lig protein levels were reduced [20]. Moreover, the heterologous expression of LigA and LigB proteins enhanced the survival of *L. bifexa* in human serum compared to the wild-type strain [21].

Lig proteins were pinpointed as the serodiagnostic markers for acute leptospirosis, and the development of immunodiagnostic assays based on Lig proteins would address the under-reporting of this neglected disease [22]. Even though *ligA* and *ligC* appeared to be present in a limited number of pathogenic serovars, the *ligB* gene was distributed ubiquitously among all pathogenic strains [23]. As the N-terminal 630 amino acids of LigA and LigB (LigCon), covering the first 6 1/2 Ig-like domains, are highly conserved between the two Lig proteins, this implies that this region should be ubiquitously present among all the pathogenic leptospiral strains. Moreover, it has been demonstrated in a Kinetic ELISA that recombinant antigen to the N terminal conserved region of LigA and LigB (rLigA/BCon) has DIVA potential and can clearly differentiate between sera of vaccinated and naturally infected dogs [24]. Hence, in silico characterization of a truncated N-terminal conserved region of Lig A and B protein comprising of first to fifth Big tandem repeat domains (rLigA/BCon1-5) of *Leptospira interrogans* serovar strain Pomona was performed in this study using a battery of bioinformatics tools to predict the feasibility of this antigen to serve as an immunodiagnostic candidate for serodiagnosis of porcine leptospirosis.

I-TASSER (Iterative Threading ASSEmbly Refinement) is an integrated platform for automated protein structure prediction that generates three-dimensional structure model of protein molecules from amino acid sequences by using structure templates from the Protein Data Bankby a technique known as fold recognition or threading [25]. Modified replica-exchange Monte Carlo simulation technique is used in I-TASSER to reassemble structural fragments from threading templates to construct full-length structure models [26]. The I-TASSER algorithm, which consists of three consecutive steps of threading, fragment assembly and iteration [27], has been employed in this study to predict the three-dimensional structure of the conserved region of LigB protein (rLigA/BCon1-5), which contains first to fifth Big tandem repeat domains spanning 400 amino acids.

In recent years, enrichment of proteome data has enabled us to predict and determine suitable regions of a protein which can act as effective immune stimulators. Various in silico methods such as ElliPro and DiscoTope, which are based on refined algorithms are available to predict as well as design diagnostic candidates from the vast proteome of pathogens. ElliPro is a structure-based webtool that implements a modified version of Thornton’s method [28] and together with a residue clustering algorithm, the MODELLER program [29,30] and the Jmol viewer, allows for the prediction and visualization of antibody epitopes in protein sequences and structures [31]. The DiscoTope server predicts discontinuous B-cell epitopes from protein three-dimensional structures and this method is based on amino acid propensities, spatial distribution, surface accessibility, and inter-molecular contacts in a compiled data set of discontinuous epitopes determined by X-ray crystallography of antibody/antigen protein complexes [32].

These computational approaches have been used in the present study to characterize the target LigA/BCon1-5 antigen. The results obtained using various in silico methods (dry lab experiments) are then compared with wet lab experiments comprising of both primary (Dot ELISA dipstick test) and secondary binding assays (Latex Agglutination Test) in order to screen porcine sera samples for leptospirosis in India.

## 2. Material and Methods

### 2.1. Serum Sample Collection and Processing

A total of 1153 serum samples from unvaccinated pigs were collected from three states of India, namely Uttar Pradesh, Maharashra and Odisha, during exsanguinations at the time of slaughtering of these animals. As there have been no previous reports of porcine leptospirosis available from the states of Maharashtra, Uttar Pradesh, and Odisha, a sample size of 384, 385 and 384 would represent a Confidence Interval of 95% and margin of error of 5% according to formula for calculation of sample size for finite population [33].A volume of 5 mL of blood was drawn into serum collection tubes (Becton Dickinson, Franklin Lakes, NJ, USA) and the blood was allowed to clot for 1 h followed by centrifugation at 2000× *g* for 10 min. After centrifugation, the serum samples were stored at −20 °C for further laboratorial analysis.

### 2.2. Leptospiral Serovars and Strains Used in Microscopic Agglutination Test (MAT)

A battery of 16 leptospiral serovars namely, *Leptospira interrogans* serovar Australis strain Ballico, *L. interrogans* serovar Autumnalis strain Akiyami A, *L. interrogans* serovar Ballum strain S102, *L. interrogans* serovar Bataviae strain vanTienen, *L. interrogans* serovar Canicola strain Hond Utrecht IV, *L. kirschneri* serovar Cynopteri strain 3522C, *L. interrogans* serovar Djasiman strain Djasiman, *L. kirschneri* serovar Grippotyphosa strain Moskva V, *L. borgpetersenii* serovar Hardjo strain Hardjoprajitno, *L. interrogans* serovarHebdomadis strain Hebdomadis, *L. interrogans* serovar Icterohaemorrhagiae strain RGA, *L. borgpetersenii* serovar Javanica strain Veldrat Batavia 46, *L. noguchii* serovar Louisiana strain LSU 1945, *L. interrogans* serovar Pomona strain Pomona, *L. interrogans* serovar Pyrogenes strain Salinem and *L. borgpetersenii* serovar Tarassovi strain Perepelitsin were employed for performing MAT.

The method for propagation of leptospiral serovars used in MAT in this study involves a periodic sub-culture on a weekly basis of 0.5 mL of week old leptospiral culture into 5 mL of freshly prepared Ellinghausen–McCullough–Johnson–Harris (EMJH) media (Difco, Sparks, MD, USA) and incubated at 29 °C in a stationary BOD incubator. The periodic sub-culturing was done in a Bio-safety Level II cabinet with a burnout facility while wearing personal protective equipment (PPE). The growth was assessed by the microscopic examination of *Leptospira* cultures for viability under dark field microscopy. Semi-solid EMJH medium containing 0.1% bacteriological agar was used to preserve leptospiral serovars used in MAT for a few months. Long-term storage of leptospiral serovars was achieved by storing leptospiral serovars in liquid nitrogen with the addition of 10% glycerol as cryoprotective agent.

### 2.3. Microscopic Agglutination Test (MAT)

MAT was employed for screening of agglutinins against various leptospiral serovars in porcine sera using the standard protocol [34]. Briefly, serum samples were diluted 1:50 in phosphate buffer saline (PBS) and a volume of leptospiral antigen, equal to the diluted serum volume, is added to each well, making the final serum dilution 1:100 in the screening test. Four-to-eight-day-old live leptospiral antigens (approx. 2 × 10^8^ leptospires/mL) of 16 reference serovars were used in this study. The microtitre plates are incubated for 2 h at 29 °C and the serum-antigen mixtures were examined using dark field microscopy. A positive outcome in MAT suggestive of exposure/seropositivity was defined as any single serum sample found to have >50% reduction in the number of free non-agglutinable leptospires in the test when compared to the control at 1:100 serum dilution for at least one leptospiral serovar. If a porcine serum sample tested positive for leptospirosis, a second incubation with twofold serial dilutions of the positive sera should be performed with specific leptospira serovars and the endpoint titre of the positive sera should be determined.

### 2.4. Prediction of the Three-Dimensional (3D) Conformation of LigA/BCon1-5 Protein

The three-dimensional structure of LigA/BCon1-5 protein (sequence from position K34 to S433, 400 amino acids) was predicted by submitting the protein sequence in I-TASSER (https://zhanglab.ccmb.med.umich.edu/I-TASSER/, accessed on 12 June 2021) server for protein homology modeling as described previously [35]. Structures of proteins such as Invasin of *Yersinia pseudotuberculosis* (PDB ID: 1CWV), Sap S-layer assembly domain of *Bacillus anthracis* (PDB ID: 6HHU) and S-layer protein rSbsC of *Geobacillus stearothermophilus* (PDB ID: 4UIC) were the various templates used during threading for the conformational annotation. The structure with minimum score was selected for further analysis. The predicted structure was obtained in the form of a protein database (pdb) compatible file. 

### 2.5. Assessment and Validation of the Predicted Structure of LigA/BCon1-5 Antigen

The structure was visualized by Chimera software and residues were marked by their single letter codes. The quality of the predicted structure was checked by ProSA (https://prosa.services.came.sbg.ac.at/prosa.php, accessed on 14 June 2021) [36], which gives an estimate of the local quality of the predicted model and a score (*z*-score), which shows the deviation from the Nuclear Magnetic Resonance (NMR) and X-ray Diffraction (XRD)-derived models. ProSA-web *z*-score plot shows only chains with less than 1000 residues and a *z*-score ≤ 10. The *z*-score of rLigA/BCon1-5 protein is highlighted as a large dot. The predicted structure was also assessed further for its veracity using SAVES v6.0 applets like ERRAT, VERIFY 3D, PROVE, PROCHECK and WHATCHECK (https://servicesn.mbi.ucla.edu/SAVES/, accessed on 14 June 2021) [37,38,39,40]. Attributes to the secondary structure was determined along with the residue distribution, which generated the Ramachandran’s plot for the prediction. The Ramachandran’s plot was predicted by SAVES v6.0.

### 2.6. Prediction of Linear and Conformational (Discontinuous) B-Cell Epitopes Present on LigA/BCon1-5 Protein

B-cell epitopes can be predicted as linear or conformational epitopes (discontinuous). The Bepipred 2.0 linear epitope prediction module was used to predict the continuous amino acid stretches which can act as a potential peptide to bind antibodies. The Ellipro module, available on the IEDB server, was used to predict both linear and conformational B-cell-epitopes present on LigA/BCon1-5 protein. Threshold was fixed to 0.5. Discontinuous epitopes were predicted by another module, DiscoTope v2.0 [41]. 

### 2.7. Prediction of T-Cell Epitopes (MHC Class I and II Peptides) Present on LigA/BCon1-5 Protein

T-cell epitopes were predicted as MHC-I and II peptides. NetMHCpan 4.1 prediction method [42] was used for MHC-I peptides. Query was made either in the form of FASTA or .pdb (protein data base) format. Swine specific alleles, SLA-1*0101, SLA-2*0101, SLA-3*0101, SLA-6*0101 were selected for the analysis. Length of the peptide was restricted to 10. MHC-II peptides were predicted using IEDB recommended 2.22 method (http://tools.iedb.org/mhcii/, accessed on 15 June 2021). Due to unavailability of swine specific locus, Human HLA-DR was selected for prediction with alleles DPA1*01/DPB1*04:01, DQA1*01:01/DQB1*02:01, DRB1*01:01. Peptide length of 15 was selected. 

### 2.8. Induction of Expression and Purification of Recombinant LigA/BCon1-5 Antigen

*Escherichia coli* (*E. coli*) M15 strain harbouring recombinant plasmid pQE30-LigA/BCon1-5 was used for rLigA/BCon1-5 antigen production as described previously [43]. *E. coli* M15 strain was grown in Luria Bertani broth (Difco) till the spectrometric reading at OD_600nm_ reached 0.5–0.7. The cells were then induced with 1 mM Isopropyl β-D-1-thiogalactopyranoside (IPTG) (Sigma-Aldrich, St. Louis, MO, USA) and allowed to grow further for 6 h at 37 °C. Cells were harvested and the proteins analyzed by Sodium dodecyl sulfate–polyacrylamidegelelectrophoresis (SDS-PAGE) using standard protocol [44]. The expected molecular weight of recombinant LigBCon1-5 fusion protein was approximately 44 kDa. Purification of recombinant protein was done using nickel–nitrilotriaceticacid (Ni-NTA) agarose affinity chromatography (Qiagen, Hilden, North Rhine-Westphalia, Germany) as per instructions of the manufacturer and analysis of purified recombinant protein was done on SDS-PAGE. Extraction of the insoluble rLigA/BCon1-5 protein from inclusion bodies was done using the denaturing agent 8M Urea.

### 2.9. Western Blot Analysis

The purified recombinant LigA/BCon1-5 antigen was checked for immunogenicity using high titred porcine sera by Western blot analysis as per standard protocol [45]. Briefly, western blot was performed by separating the purified recombinant protein on 12% SDS-PAGE and transferring it onto nitrocellulose membrane (NCM) (Thermo Fisher, Waltham, MA, USA) using transfer buffer by applying current at 0.8 mA/cm^2^ in a blotting apparatus (BioRad, Hercules, CA, USA). Successful transfer of the protein was confirmed by staining the membranes with the Ponceau S stain. After overnight blocking with 3% skimmed milk, LigA/BCon1-5 protein present on the membrane was immune-stained by first exposing the membrane to positive sera control—a pig sera—which showed MAT titre of 1:3200 against *L. Pomona* (received from Tamil Nadu Veterinary and Animal Sciences University (TANUVAS), Chennai, India) and known negative porcine sera confirmed by MAT at a dilution of 1:50 followed by species-specific secondary antibodies conjugated to horse radish peroxidase at a dilution of 1:5000. Finally, the reaction was developed with either 4-chloro-1-naphthol (Sigma-Aldrich) or 3, 3′- Diamino benzidine (DAB) (Sigma-Aldrich). In order to ascertain whether field sera having high MAT titre against other leptospiral serovars such as *L. Australis*, *L. icterohaemorrhagiae*, *L. grippotyphosa* and *L. Tarassovi* react with rLigA/BCon1-5 antigen obtained by expressing LigB gene of *L. Pomona*, we also employed porcine field sera reactive to these serovars in western blot.

### 2.10. Recombinant LigA/BCon1-5 Based Latex Agglutination Test (rLigA/BCon1-5 Based LAT)

Latex beads were sensitized with rLigA/BCon1-5 antigen as described previously [46] with slight modifications. A 10% suspension of latex particles (0.8 µm diameter, Sigma-Aldrich) was washed thrice with glycine buffered saline (Glycine 0.1 M, NaCl 0.17 M; pH 8.2). Finally, the latex beads were made into a 2% suspension with glycine buffered saline which was later mixed with an equal volume of rLigA/BCon1-5 antigen (1mg/mL) diluted in the same buffer. The mixture was incubated at 37 °C for 6 h in a shaking platform to ensure constant mixing. The sensitized latex beads were further blocked with Bovine Serum Albumin (Difco) (5mg/mL) and incubated overnight. Latex beads were centrifuged, and the pellet was finally resuspended in glycine-buffered saline as a 2% suspension containing 0.02% sodium azide. The sensitized latex beads were stored at 4 °C until use. LAT was performed on glass slides by mixing equal volume of serum (20 µL) and sensitized beads (20 µL). The result was read within 2 min. Samples were considered positive when there is formation of agglutination. A score of 3+, 2+ and 1+ were designated to sera which showed agglutination within 30 s, 30 s–1 min, 1–2 min, respectively, and as described previously [47]. Samples were considered negative if no agglutination was observed. 

### 2.11. Recombinant LigA/BCon1-5 Based IgG Dot-ELISA Dipstick Test

IgG Dot-ELISA was carried out based on the method described previously [11] with minor modifications. IgG Dot-ELISA was standardized by dotting nitrocellulose membrane (NCM) (Thermo Fisher, Waltham, MA, USA) provided at the tips of plastic combs with 2 µL of rLigA/BCon1-5 antigen in carbonate-bicarbonate buffer in various concentrations (25 ng to 1 µg) and allowed to be air dried for 1 h at 37 °C. The unoccupied sites of the NCM were then blocked with 5% skim milk powder in PBS-T (0.05% Tween-20 in PBS) overnight at 4 °C. These coated NCM were washed with washing buffer comprising of 0.05% Tween-20 in PBS. Various dilutions of porcine sera, starting from 1:50 to 1:800, were diluted in blocking buffer and dispensed into wells of microtitre plate (200 µL/well). The antigen coated combs were dipped in the wells of the micro titre plate and incubated for 1h at 37 °C. Combs were then washed with washing buffer three times, each for duration of five minutes. After washing, the combs were dispensed in various dilutions (2000–10,000) of rabbit anti-pig IgG HRPO conjugate (Sigma-Aldrich) in blocking buffer and incubated for 1h at 37 °C. The combs were then washed thrice with PBS-T and developed with 200 µL of substrate buffer containing 3, 3′-Diamino benzidine (DAB) (Sigma-Aldrich) for 15 min in dark. The reaction was stopped by rinsing the strips with PBS. Appearances of brown colour dots on NCM indicate positive reaction. Non-appearance of brown colour dot on the nitrocellulose membrane indicated negative result. All of the samples were tested by including known positive and negative control.

### 2.12. Statistical Analysis

The relative sensitivity, specificity and accuracy (in percentage) of rLigA/BCon1-5 based Dot ELISA Dipstick test and LAT in comparison to MAT for the serodiagnosis of porcine leptospirosis was described below

Sensitivity = a/(a + c) × 100, where ‘a’ is the number of serum samples positive by both MAT and rLigA/BCon1-5 based Dot ELISA Dipstick test/LAT, ‘c’ is the number of serum samples positive by MAT but negative by rLigA/BCon1-5 based Dot ELISA Dipstick test/LAT.

Specificity = d/(b + d) × 100 where ‘d’ is the number of serum samples negative by both MAT and rLigA/BCon1-5 based Dot ELISA Dipstick test/LAT, ‘b’ the number of serum samples negative by MAT but positive by rLigA/BCon1-5based Dot ELISA Dipstick test/LAT.Accuracy = a + d/(a + b + c + d) × 100

An intuitive method for calculating predictive values (in percent) for positive and negative test results was done as per standard method [48].Positive Predictive Value (PPV) = a/(a + b) × 100
Negative Predictive Value (NPV) = d/(c + d) × 100

The evaluation of rLigA/BCon1-5 based Dot ELISA Dipstick test/LAT for detection of anti-*Leptospira* antibodies in pigs as compared with MAT was determined using Kappa statistics as described previously [43].

## 3. Results

### 3.1. Microscopic Agglutination Test

The MAT results revealed a seropositivity of 23.85% (275/1153) for porcine leptospirosis with maximum agglutinins detected for serovar Icterohaemorrhagiae (n = 143) followed by Grippotyphosa (n = 97), Pomona (n = 50), Tarassovi (n = 37), Javanica (n = 18), Australis (n = 06) and Pyrogenes (n = 06) as shown in Figure 1.The cumulative figure of sera positive for various serovars (N = 357) exceeded thetotal MAT positive sera (n = 275) as several sera reacted with multiple leptospiral serovars. The MAT titre ranged from 1in 100 to 1in 3200. 

A total of 384, 385 and 384 sera were received from Maharashtra, Uttar Pradesh and Odisha, respectively, of which 104, 92 and 79 sera tested positive by MAT, respectively. Thus, Maharashtra recorded the highest seropositivity of 27.08% (104/384) followed by Uttar Pradesh (23.89%) (92/385) and Odisha (20.57%) (79/384). Icterohaemorrhagiae was the leading serovar recorded in all the three Indian states. The serovar distribution in all the three Indian states is depicted in Figure 1.

### 3.2. Three-Dimensional (3D) Conformation of LigA/BCon1-5 Protein

Thepredicted three-dimensional structure of LigA/BCon1-5 protein visualized using Chimera software (v1.13.1, University of California, San Francisco, CA, USA) is depicted in Figure 2A. Based on MolProbity scores, 90.71% of amino acids were Ramachandran favoured (i.e., without any steric clashes) and 3.01% were Ramachandran outliers (Figure 2B). The majority of the amino acid residues present in the protein may adopt β-sheet conformation as seen in this plot (Figure 2B).

### 3.3. Assessment and Validation of the Predicted Structure of LigA/BCon1-5 Antigen

The local quality score of the predicted structure of LigA/BCon1-5 antigen (Figure 2C) indicates that the predicted structure is reliable. The *z*-score of −4.97 for the predicted structure calculated using ProSA (Figure 2D) is well within the range of native conformations for proteins of similar size and this indicates that the predicted structure is reliable. Ramachandran plot of the protein structure predicted by SAVES v6.0 to analyze the stereo-chemical quality of the model showed that only 3.01% of amino acids were Ramachandran outliers suggesting that the predicted structure is reliable.

### 3.4. Prediction of Linear and Conformational (Discontinuous) B-Cell Epitopes Present on LigA/BCon1-5 Protein

The graphical representation of amino acid scores as their propensity to be a part of linear B-cell epitope is depicted in Figure 3A, while Figure 3B is the graphical representation of amino acid scores as their propensity to be part of conformational (discontinuous) B-cell epitope. The amino acids with scores above the threshold value (default value is 0.5) according to Bepipred Linear Epitope Prediction 2.0 module are highlighted in yellow (Figure 3A) while the amino acids with scores above the threshold value (default value is −3.7) according to the DiscoTope Prediction server are highlighted in green (Figure 3B). 

The ElliPro prediction server suggested 10 and three potential linear and conformational B-cell-epitopes, respectively, on the LigA/BCon1-5 protein (Figure 4 and Figure 5). The peptide sequence and number of amino acid residues present in linear and conformational B-cell-epitopes and their relative position on the protein as well as their score are depicted in Figure 4 and Figure 5, respectively. Furthermore, the pictorial presentation (ball and stick model) of linear and conformational B-cell-epitopes (highlighted in yellow) on the peptide backbone is also shown in Figure 4 and Figure 5, respectively. The linear B-cell-epitopes ranged from 4–47 amino acids in length while the three potential conformational B-cell-epitopes comprised of 47, 61 and 89 amino acids, respectively.

The DiscoTope prediction server suggested 47 amino acid residues (based on attributes such as surface accessibility which is estimated in terms of contact numbers, and a novel epitope propensity amino acid score) to be part of the B-cell antigen (Table 1). The amino acid residues, their position on protein backbone, contact number and propensity scores are shown in Table 1. The final DiscoTope score (obtained by combining the propensity scores of amino acid residues in spatial proximity and the contact numbers) is also depicted in Table 1. The pictorial presentation of a few important amino acid residues which form part of the discontinuous epitope as predicted by DiscoTope server is shown on the peptide backbone in Figure 5D.

### 3.5. Prediction of T-Cell Epitopes (MHC Class I and IIPeptides) Present on LigA/BCon1-5 Protein

NetMHCpan 4.1 prediction server and IEDB recommended 2.22 method predicted that there were several amino acid stretches distributed across the whole length of LigA/BCon1-5 protein that may fit to MHC-I and MHC-II grooves. Ten of the most efficient peptides for MHC-I and II grooves are presented with their scores in Table 2 and Table 3, respectively. Interestingly, sequences such as QSVVTI, GSVKL and STDFKVTQAA (highlighted in blue, yellow and green shades in Table 2 and Table 3) can fit and can be presented by both the MHCs. 

### 3.6. Recombinant LigA/BCon1-5 Antigen Expression

In this study, a high-level expression of the rLigA/BCon1-5 protein of about 20 mg/L of purified recombinant protein was obtained (Figure 6A). Even though the expected molecular weight of the protein was 44 kDa, the observed molecular weight of the protein on SDS-PAGE was approximately 51 kDa. The expression kinetics of rLigA/BCon1-5 protein revealed that the protein first appeared on SDS-PAGE 2 h after IPTG induction and reached maximum level at 8-h post induction (data not shown). 

### 3.7. Western Blot Analysis

Western blot analysis showed that rLigA/BCon1-5 protein was highly immunogenic against anti-*Leptospira* antibodies present in swine sera since clearly visible bands of either brown or purple colour were obtained when either 3,3′-Di amino benzidine (DAB) or 4– choloro-1– naphthol were used as substrate for HRP conjugated secondary antibodies (Figure 6B).Further, we have found that rLigBCon1-5 antigen reacted in western blot to field sera seropositive for *L. Pomona* (Positive control) as well as with several other leptospiral serovars (*L. australis, L. icterohaemorrhagiae*, *L. grippotyphosa* and *L. tarassovi*), which suggested the ubiquitous presence of LigB protein in all pathogenic leptospiral serovars.

### 3.8. Recombinant LigA/BCon1-5 Based IgG Dot-ELISA Dipstick Test

TheIgG-based Dot ELISA Dipstick testwas standardized by dotting NCM provided at the tip of each plastic combs with 1µg of purified rLigA/BCon1-5 antigen and the optimum dilutions for field sera and anti-pig IgG HRP conjugate were found to be 1:50 and 1:2500, respectively. The field sera, which gave brown colour dots on NCM, were considered as positive while non-reactors failed to give brown colour dot on NCM (Figure 7). Of the 1153 sera sample tested, 275 (23.85%) and 269 (23.33%) sera samples were found positive by MAT and IgG Dot-ELISA Dipstick test respectively. The sensitivity, specificity, accuracy and Kappa value of rLigA/BCon1-5 based IgG Dot ELISA Dipstick test relative to MAT is shown in Figure 7.

### 3.9. Latex Agglutination Test and Correlation between MAT Titre and LAT Score

The Latex agglutination test (LAT) showed a clear-cut agglutination with positive sera (Figure 8D–F) which can be visualized easily and can clearly differentiate negative sera showing homogeneous suspension (Figure 8A–C). There exists a positive correlation between MAT titre and LAT score. Porcine sera (n = 65) which gave MAT titre ≥1:400 gave LAT score of 3+, which meant that a high intensity of agglutination occurred almost instantaneously (usually <30 s) upon addition of serum samples which is clearly visible in the form of tiny flakes (Figure 8F). Sera samples (n = 67) with a MAT titre 1:200 gave LAT score of 2+ and indicated that the agglutination occurred between 30s–1min, respectively, with moderate intensity of agglutination in which the agglutinins formed a halo at the periphery during the process of swirling of the glass slide while the centre was virtually empty (Figure 8E). Sera samples (n = 112) with a MAT titre of 1:100 gave LAT score of 1+ and agglutination occurred in 2 min with low intensity of agglutination (halo at periphery not prominent) (Figure 8D). Thirty-one MAT^+ve^ sera (1: 100 titre) gave no agglutination in LAT, while nine MAT^−ve^ sera gave 1+ LAT score. Of the 1153 sera samples tested, 275 (23.85%) and 244 (21.16%) sera samples were found positive by MAT and LAT, respectively. The sensitivity, specificity, accuracy and Kappa value of rLigA/BCon1-5 based LAT relative to MAT are shown in Figure 8. 

## 4. Discussion

Porcine leptospirosis is a major public health problem since pigs harbour various leptospiral serovars in the kidneys and genital tracts with intermittent shedding of the organisms to the environment through urine and genital discharges [49]. In humans involved in swine husbandry, leptospirosis is known as “swine herder’s disease”, which suggests the zoonotic nature of this disease [50]. The fact that leptospirosis is a major health problem in porcine population along with their role as a potential reservoir for cross species disease transmission to humans makes early identification and management of this disease of paramount importance in swine [51]. Moreover, antibiotictreatmentismosteffectivewheninitiatedearlyinthecourseofthedisease [52]. Therefore, the current need should be focused on the development of penside diagnostics such as the Dot ELISA Dipstick test and Latex Agglutination Test (LAT), which can aid in diagnosing the disease at field level itself without resorting for time consuming laboratory tests such as MAT which can delay disease diagnosis [53].

Icterohaemorrhagiae was the predominant serovar (52.0%, 143/275) observed in this study. Besides India, in developing economies such as Brazil, Icterohaemorrhagiae is gradually replacing Pomona as the predominant leptospiral serovar in swine herds, which suggests rodent control failure in pig farms [54,55] as well as failure to comply with the principles of external biosecurity rules. This is in stark contrast to the leptospiral serovar epidemiology observed in swine in Europe, which is dominated by leptospiral serovars present in serogroups such as Sejroe [1], Pomona [4,5] and Australis [4,6]. 

It is also noteworthy to observe that pig farms that did not perform strategic control of rodents presented 7.8 times higher chance of infection with leptospirosis [56]. The N-terminal 630 amino acids of LigA and LigB (LigCon), covering the first 6 1/2 Ig-like domains, are highly conserved between the two Lig proteins [57] and this region holds promise for developing DIVA capability diagnostic assays [24]. NMR solution structure for LigBCen2R, which is a partial eleventh and full twelfth bacterial immunoglobulin-like domain of LigB (PDB ID: 2MQG) [58] and LigB-12, which is the twelfth and most C-terminal Ig-like domain from LigB (PDB ID: 2MOG) were available in the PDB structure database [57]. Further, a low-resolution solution structure of the main immunoreactiveregion of the LigB protein, comprising of a series of 12 connected ‘immunoglobulin-like’ domain regions, was made available using small angle X-ray scattering (SAXS).The results obtained using SAXS showed that this region was highly elongatedand the extended arrangement with notable bends of this immunoreactive region of LigB protein encouraged the exploration of highly-exposed surface for immunoreactivity using a library of anti-LigB monoclonal antibodies (mAbs) [59].

The Ramachandran plot gives information regarding energetically allowed and forbidden regions for the dihedral angles [60]. The distribution of the Phi/Psi values observed in a protein structure can be used for structure validation [61]. For poor quality homology models, many dihedral angles will be found in the forbidden regions of the Ramachandran plot and such deviations usually indicate problems with the predicted structure [60]. In this study, the three-dimensional structure predicted for LigA/BCon1-5 protein was highly reliable since only 3.01% were Ramachandran outliers.

ProSA-web *z*-score plot shows *z*-scores of all protein chains in PDB determined by X-ray crystallography or NMR spectroscopy with respect to their length. The *z*-score indicates overall quality of a predicted structure and measures the deviation of the total energy of the predicted structure in comparison to an energy distribution derived from random conformations [62]. *Z*-scores that deviate strongly from the data base average are unusual and such structures frequently turn out to be erroneous [36]. In the case of rLigA/BCon1-5 protein, the Z score of −4.97 falls well within the range of native conformations for proteins of similar size which indicates that the predicted structure is reliable.

Insilicoprediction that is conducted using a battery of immunoinformatic serversfor the prediction of linear and conformational B-cell-epitopes such as BepiPred-2.0, ElliPro and DiscoTope prediction servers proved to be highly reliable with regard to predicting the immunogenicity of rLigA/BCon1-5 antigen since the primary (IgG Dot ELISA Dipstick test) and secondary binding assays (LAT) developed using this recombinant protein showed high sensitivity and specificity with gold standard test, MAT. The web-based B-cell epitope prediction tool, ElliPro, treats individual protein as an ellipsoid and calculates protrusion indexes for protein residues protruding from the globular surface of the protein which are available for interaction with antibodies [31]. The advantage of in silicoprediction methods are their relatively low-cost and high-throughput, which are significantadvantagesin using them topredictboth B- and T-cell epitopes, as well as assist in the selection of leadproteins(based onpredicted immunogenicity) from the vast proteome of bacterial pathogens such as *Leptospira interrogans*. 

The present study with regard to the immunogenicity of rLigA/BCon1-5 antigen is in concordance with the findings of a previous study in which two sets of hybridoma cell lines were generated for mAb production against two LigB truncations, LigB1-7 and LigB7-12 in order to studyhost immune response against the twelve ‘immunoglobulin-like’ domain region of LigB protein [59]. Based on dissociation constants (K_D_ values) for each of these mAbs, it was observed that mAbs from LigB1-7 antigen-derived library were able to bind tighter to the LigB antigen than mAbs from LigB7-12 antigen-derived librarysuggesting that the anterior portion of LigB protein chosen in the present study was more immunogenic than the posterior portion [59]. The domain-level specificity of individual LigB mAbs within LigB1-7 antigen-derived mAb library investigated using a comprehensive set of single as well as double Ig-like domain LigB truncates showed that immunoreactivity of mAbs was weighted towards LigB1-2 and LigB4-5 regions which were Ig-like domains found within rLigBCon1-5 antigen chosen in the present study [59]. Only one single domain, LigB3, and one double domain, LigB3-4, lacked immunogenicity within recombinant LigBCon1-5 antigen for the set of nine mAbs within LigB1-7 antigen-derived library [59].

The NetMHCpan 4.1 prediction server is one of the most reliable prediction servers that has been trained for 75 different swine MHC Class I alleles based on an artificial neural network algorithm. Further, the NetMHCpan 4.1 prediction server is also trained on a combination of more than 850,000 quantitative binding affinity (BA) and Mass-spectrometry Eluted Ligands (EL) peptides. Interestingly, three out of ten of the most efficient peptides for MHC-I grooves predicted by this server in the present study can also fit and be presented by MHC Class II molecules, indicating that both CMI and humoral immune response can be induced by these peptides.

The binding of T cell receptor (TCR) to a peptide complexed with major histocompatibility complex (MHC) class II molecule on antigen presenting cells (APCs) results in the activation of naive T_H_ cells. The activated effector T_H_ cells secrete cytokines which is a prerequisite for efficient antibody response. Hence, in order to predict MHC class II restricted peptides, the protein sequences were submitted to IEDB recommended 2.22 method. As the server could not accommodate swine MHC Class II isotypes, we have used human MHC Class II isotypes for the prediction in our study since leptospirosis is a common disease of both swine and humans.

The main limitation of pan-specific or cross species approach for MHC class II prediction is that there are considerable differences in sequence polymorphism and corresponding details in the molecular structures across the different MHC class II loci which complicate the development of cross-species training strategies [63]. This, combined with the very limited amount of data available for most MHC class II molecules, has limited the application of pan-specific methods to HLA-DR molecules which was used in the present study to extrapolate swine MHC-II binding peptides due to unavailability of swine specific MHC Class II loci in IEDB recommended 2.22 method. Like IEDB, MHCpred, SYFPEITHI and most of the available prediction servers have this major constraint. Fortunately, most proteins of the porcine immune system share structural and functional similarities with their human counterparts, and it is a noteworthy observation that the porcine immune system closely resembles humans for >80% of analyzed parameters [64]. Therefore, it was planned to rely on the MHC-II processing of human allelomorphs. Further, the antigens predicted by these servers are based upon a pattern of hydrophilicity index and different other biochemical parameters. So, there is every possibility that the peptides and their amino acids predicted by human MHC-II allele may be spanning the exact peptides of swine leukocyte antigens.

The rLigA/BCon1-5 based IgG Dot ELISA Dipstick testgave results in concordance to MAT in terms of sensitivity (97.82%), specificity (94.99%) and accuracy (95.66%). In this study, only six porcine serum of 1:100 MAT titre gave negative test result with IgG-based Dot-ELISA Dipstick test. This indicates that IgG-based Dot-ELISA Dipstick test is highly sensitive in comparison to MAT. However, 44 serum samples which tested positive by IgG-based Dot-ELISA Dipstick test, gave negative test result by MAT. The probable reason for the slightly lower specificity (94.99%) of IgG-based Dot-ELISA Dipstick test might be that all the sera construed as MAT negative might not actually be negative, since the battery of leptospiral antigen used for performing MAT in this study was only sixteen *Leptospira* serovars. There are more than 250 pathogenic serovars belonging to 24 serogroups reported within the species *L. interrogans* [65]. Therefore, it makes practical sense to maintain in a laboratory only a panel of locally circulating serovars present in an endemic region and when an animal is infected with a newly emerging serovar which is not included in the panel of leptospiral serovars used for MAT, MAT invariably gives a false negative test result due to incomplete panel of leptospiral serovars [66]. Our observations on IgG based Dot-ELISA Dipstick test are congruous to the findings of other disease investigators [11,12], who also reported high sensitivity and specificity while employing IgG based Dot-ELISA Dipstick test for detection of anti-*Leptospira* antibodies.

Even though both the LAT and IgG Dot ELISA Dipstick test employed the rLigA/BCon1-5 antigen, LAT showed less sensitivity (88.72%) in comparison to IgG Dot ELISA Dipstick test (97.82%) when both the tests were compared with MAT. However, the specificity of 98.97% obtained for LAT was higher than that obtained for IgG Dot ELISA Dipstick test (94.99%). Moreover, LAT provides test results much faster (within two minutes) in comparison to IgG Dot ELISA Dipstick test. In a limited field trial undertaken by our research team in a resource-constrained setting using rLigA/BCon1-5 based LAT in collaboration with an ambulatory veterinary clinic in India, it was revealed that LAT assay reagents possessed long shelf life of at least three months in tropical conditions with high humidity and temperature, even with frequent electricity cuts in India affecting refrigeration temperatures, owing to the addition of 0.02% Sodium azide as preservative for sensitized latex bead suspension.

Both the LAT and IgG Dot ELISA Dipstick test are field-oriented, highly economical, spot tests which require less sophisticated equipments [10,11] which is in stark contrast to MAT which is a laboratory-oriented test, which requires costly equipments such as Dark Field Microscope (DFM) and Biological Oxygen demand (BOD) Incubator. Moreover, both LAT and IgG Dot ELISA Dipstick test are user friendly tests which can be easily performed, and test results interpretation can be done by non-skilled personnel with minimal training. This is in contrast to MAT, which is a cumbersome test due to the requirement of live leptospiral antigens whose maintenance is a tedious task which requires a highly skilled and dedicated technical staff with optimum training in handling live leptospiral cultures [9]. Moreover, both the LAT and IgG Dot ELISA Dipstick test generates limited amounts of biomedical waste and their alluring properties include their simplicity and portability [12,67], which is in stark contrast to MA,T which generates considerable biomedical waste, mostly in the form of old leptospiral cultures that requires a proper disposal system and is non-portable due to the requirement of equipment and other laboratory facilities. Hence, the LAT and IgG Dot ELISA Dipstick test are diagnostic assays that are ideally suited for use at the peripheral level of animal health care system especially in resource-constrained settings as “point of care” tests for diagnosis of porcine leptospirosis.

## 5. Conclusions and Further Perspectives

The use of immunoinformatic tools proved to be highly reliable with regard to predicting the immunogenicity of recombinant LigA/BCon1-5 antigen which showed promise as a serodiagnostic marker when employed in primary (IgG Dot ELISA Dipstick test) and secondary binding assays (LAT) for detection of leptospirosis. Immunoinformatics approach for the prediction of linear and conformational B-cell-epitopes such as the BepiPred-2.0, ElliPro and DiscoTope prediction servers can assist in pinpointing potential immunogenicprotein candidatesfrom the vast proteome of bacterial pathogens. The use of recombinant LigA/BCon1-5 protein-based spot tests such as the Dot-ELISA Dipstick test and LAT as rapid-screening tests in remote locations, where MAT is not readily accessible, would permit the implementation of intervention strategies based on early case detection and the timely initiation of antimicrobial therapy, which would prevent disease progression and the severe outcomes associated with leptospirosis.

## Figures and Tables

**Figure 1 pathogens-10-01082-f001:**
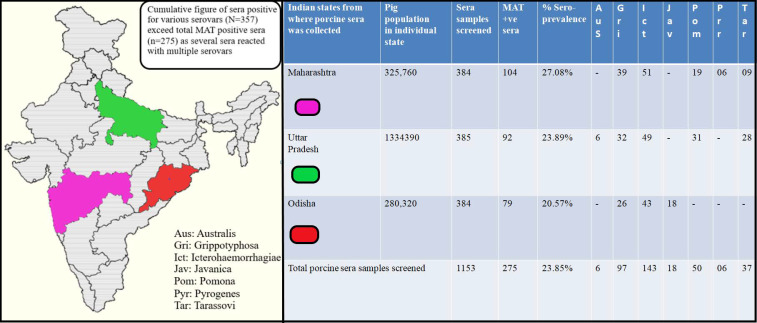
Map of India showing locations from where porcine serum samples were collected for performing Microscopic Agglutination Test (MAT). The porcine sera showing agglutinins for at least one leptospiral serovar at 1:100 cut off titre is considered to be MAT positive.

**Figure 2 pathogens-10-01082-f002:**
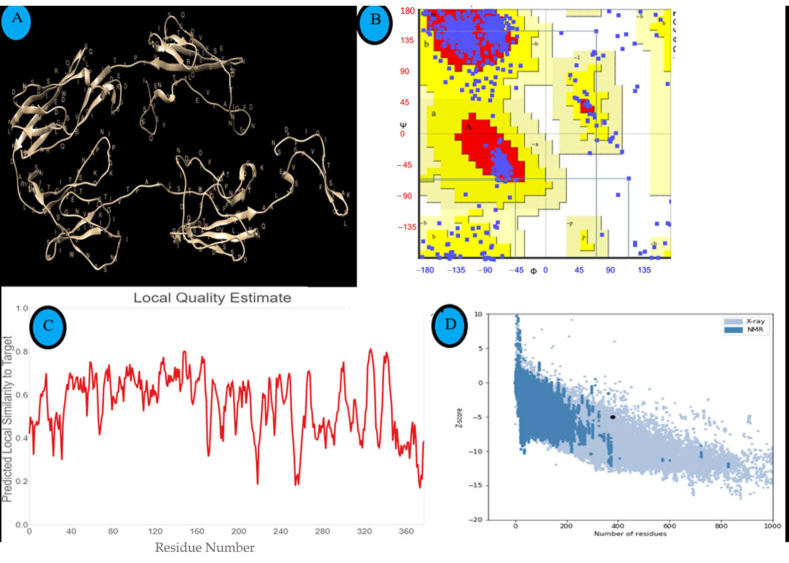
3D-structure of LigA/BCon1-5 protein and assessment of quality of predicted structure. (**A**) Predicted 3D structure of LigA/BCon1-5 protein visualized by Chimera software (v 1.13.1). (**B**) Ramachandran’s plot of LigA/BCon1-5 protein in order to access the quality of the predicted structure. (**C**) Local quality score of the predicted structure given in a graphical form where X & Y axes represents residue position & predicted local similarity to the target respectively. (**D**) Z Score of the predicted structure calculated using ProSA & found to be– 4.97. Black Dot represents submitted protein query.

**Figure 3 pathogens-10-01082-f003:**
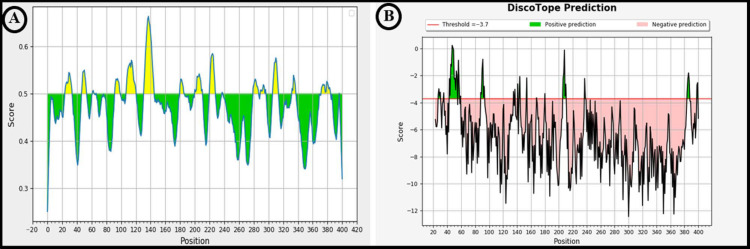
Graphical presentation of amino acid residue score as their potential to be a part of (**A**) linear epitope or (**B**) discontinuous epitope. (**A**) Amino acid residue with a score above the threshold (default value is 0.5 in Bepipred Linear Epitope Prediction 2.0) is considered to be part of an epitope & coloured in yellow. The X and Y axes represents residue position and score respectively. (**B**) The graph represents the score of amino acid residues with tendency to form a discontinuous epitope in DiscoTope Prediction Server. The threshold is– 3.7 & the positive prediction are shown in green.

**Figure 4 pathogens-10-01082-f004:**
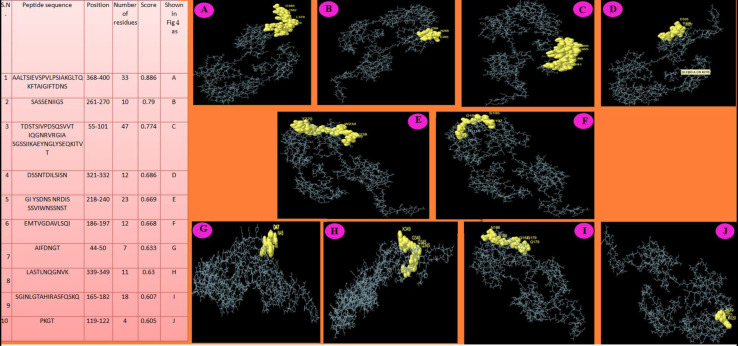
Linear B cell epitopes predicted using Ellipro prediction server. (**A**–**J**) Ball & stick model of Linear B cell epitopes present in LigA/BConl-5 protein predicted by ElliPro prediction server. Yellow highlight regions are continuous/linear stretches of amino acid residues with ability to stimulate B cells.

**Figure 5 pathogens-10-01082-f005:**
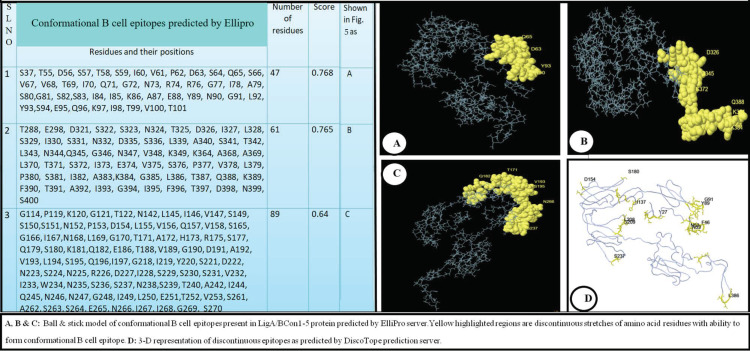
Conformational (Discontinuous B Cell Epitope prediction using ElliPro and DiscoTope prediction servers. (**A**–**C**) Ball & stick model of conformational B cell epitopes present in LigA/BConl-5 protein predicted by ElliPro server: Yellow highlighted regions are discontinuous stretches of amino acid residues with ability to form conformational B cell epitope. (**D**) 3-D representation of discontinuous epitopes as predicted by DiscoTope prediction server.

**Figure 6 pathogens-10-01082-f006:**
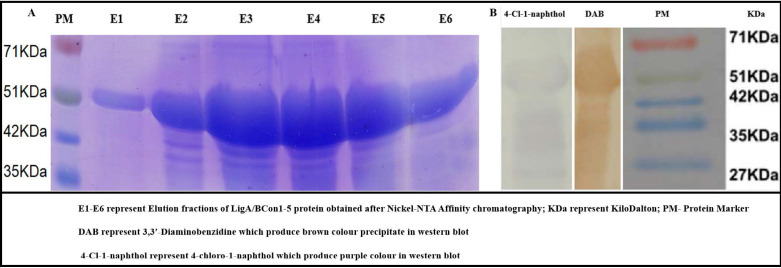
Purification and immunogenicity testing of recombinant LigA/BCon1-5 protein. (**A**) Sodium Dodecyl Sulfate Polyacrylamide Gel Electrophoresis of elution fractions of recombinant LigA/BCon1-5 protein obtained after Ni-NTA affinity chromatography. (**B**) Western blot analysis for testing immunogenicity of recombinant LigA/BCon1-5 protein. (E1–E6) represent Elution fraction of LigA/BCon1-5 protein obtained after Nickel-NTA Affinity chromatography; KDa represent Kilodalton; PM-Protein Marker DAB represents 3,3′-Diaminobenzidine which produce brown colour precipitate in western blot. 4-Cl-1-naphthol represents 4-chloro-1-naphthol which produce purple colour in western blot.

**Figure 7 pathogens-10-01082-f007:**
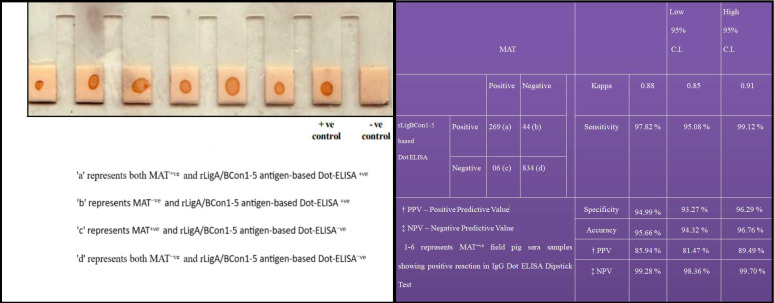
Relative sensitivity, specificity and accuracy values of recombinant LigA/BCon1-5 antigen-based IgG Dot-ELISA Dipstick Test for detection of anti-leptospiral antibodies in porcine sera as compared to MAT.

**Figure 8 pathogens-10-01082-f008:**
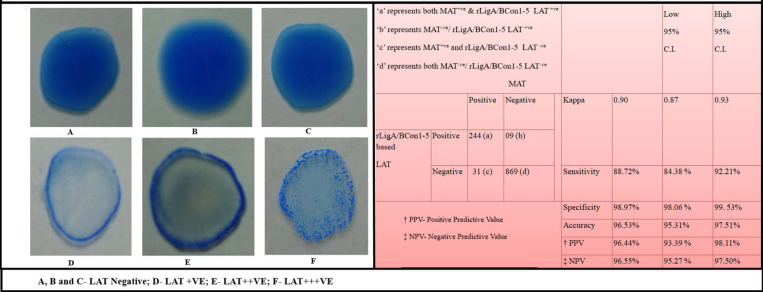
Relative sensitivity, specificity and accuracy values of recombinant LigA/BCon1-5 antigen based latex agglutination test for detection of anti-leptospiral antibodies in porcine sera as compared to Microscopic Agglutination Test.

**Table 1 pathogens-10-01082-t001:** DiscoTope predicted residues having propensity to be part of a B-cell epitope.

S.N.	Residue Position	Residue	Contact Number	Propensity Score	Discotope Score
1.	27	TYR	2	−3.591	−3.408
2.	28	GLN	3	−2.973	−2.976
3.	29	ASP	7	−3.074	−3.525
4.	30	SER	4	−3.131	−3.231
5.	35	GLY	4	−3.18	−3.274
6.	43	THR	6	−1.736	−2.227
7.	44	ALA	20	−1.33	−3.477
8.	45	ILE	4	−0.794	−1.163
9.	46	PHE	7	−0.591	−1.328
10.	47	ASP	2	0.531	0.24
11.	48	ASN	3	0.474	0.075
12.	49	GLY	1	0.042	−0.078
13.	50	THR	7	−1.634	−2.251
14.	51	ASN	7	−1.566	−2.19
15.	52	GLN	15	−1.506	−3.058
16.	53	ASN	3	−1.485	−1.659
17.	54	ILE	1	−2.099	−1.973
18.	56	ASP	0	−0.973	−0.861
19.	57	SER	13	−1.195	−2.553
20.	59	SER	8	−2.936	−3.518
21.	88	GLU	10	−2.378	−3.254
22.	89	TYR	16	−1.74	−3.38
23.	90	ASN	7	−0.73	−1.451
24.	91	GLY	1	−0.767	−0.793
25.	92	LEU	19	−1.608	−3.608
26.	93	TYR	6	−2.374	−2.791
27.	135	GLY	1	−3.453	−3.171
28.	137	HIS	12	−1.846	−3.014
29.	141	SER	6	−2.173	−2.613
30.	143	ASP	9	−2.552	−3.294
31.	144	PRO	0	−2.338	−2.069
32.	154	ASP	0	−2.449	−2.167
33.	168	ASN	8	−3.083	−3.649
34.	180	SER	7	−2.855	−3.332
35.	206	ILE	8	−1.525	−2.269
36.	207	PRO	8	−1.009	−1.813
37.	208	LEU	0	−0.125	−0.111
38.	209	GLY	14	−2.144	−3.507
39.	210	LYS	1	−2.851	−2.638
40.	237	SER	0	−2.495	−2.208
41.	238	ASN	9	−2.897	−3.599
42.	384	LYS	6	−2.736	−3.111
43.	385	GLY	0	−2.522	−2.232
44.	386	LEU	0	−2.034	−1.8
45.	387	THR	3	−2.273	−2.357
46.	398	ASP	0	−3.043	−2.693
47.	399	ASN	0	−2.853	−2.525

**Table 2 pathogens-10-01082-t002:** MHC-I binding peptides predicted by the NetMHCpan EL 4.1.

S.N.	Peptide Sequence	Allele	Position	Score
1.	VPDSQSVVTI	SLA-1*0101	61–70	0.245636
2.	IEVSPVLPSI	SLA-6*0101	373–382	0.210248
3.	HQDISNDPLI	SLA-1*0101	137–146	0.207865
4.	ETVDTGIVTI	SLA-1*0101	251–260	0.203195
5.	SGSSIIKAEY	SLA-2*0101	80–89	0.182214
6.	SIANGTSTTL	SLA-1*0101	31–40	0.178052
7.	SIAKGLTQKF	SLA-2*0101	381–390	0.163866
8.	SHQDISNDPL	SLA-1*0101	136–145	0.163175
9.	SENIIGSVKL	SLA-6*0101	264–273	0.162667
10.	STDFKVTQAA	SLA-1*0101	360–369	0.15193

**Table 3 pathogens-10-01082-t003:** MHC-II binding peptides predicted by the IEDB recommended 2.22 method.

S.N.	Peptide Sequence	Allele	Position	Adjusted Rank *	Method Used
1.	GSVKLIVTPAALVSI	HLA-DRB1*01:01	269–283	0.91	Consensus (comb.lib./smm/nn)
2.	GSTDFKVTQAALTSI	HLA-DRB1*01:01	359–373	1.10	Consensus (comb.lib./smm/nn)
3.	LSFFHLLLGNSNPTI	HLA-DRB1*01:01	6–20	1.30	Consensus (comb.lib./smm/nn)
4.	QSVVTIQGNRVRGIA	HLA-DRB1*01:01	65–79	3.60	Consensus (comb.lib./smm/nn)
5.	FHLLLGNSNPTITRI	HLA-DRB1*01:01	9–23	5.40	Consensus (comb.lib./smm/nn)
6.	PLIVWSSSNPDLVQV	HLADQA1*01:01/DQB1*02:01	144–158	7.30	NetMHCIIpan
7.	GKKQKLIATGIYSDN	HLA-DRB1*01:01	209–223	12	Consensus (comb.lib./smm/nn)
8.	AEEMTVGDAVLSQIQ	HLADQA1*01:01/DQB1*02:01	184–198	13.00	NetMHCIIpan
9.	KKQKLIATGIYSDNS	HLA-DRB1*01:01	210–224	13.0	Consensus (comb.lib./smm/nn)
10.	NRVRGIASGSSIIKA	HLA-DRB1*01:01	73–87	14.0	Consensus (comb.lib./smm/nn)

* Lower adjusted rank indicates a better binding.

## Data Availability

The data used to support the findings of this study are included within the article. Further, the data presented in this study will be made available upon request.

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
