# Peer review of "Immunoinformatic Study of Recombinant LigA/BCon1-5 Antigen and Evaluation of Its Diagnostic Potential in Primary and Secondary Binding Tests for Serodiagnosis of Porcine Leptospirosis"

_pathogens, 2021, doi:10.3390/pathogens10091082_

Round 1

Reviewer 1 Report

Line 70 – Taking into account immunization of pigs, vaccinations are of less importance as their effectiveness is limited. Leptospirosis is mainly controlled by antibiotic therapy.

Line 162 - In chapter “2.3. Microscopic agglutination test (MAT)” authors should specify that, if a serum sample is positive for leptospirosis, a second incubation with dilution series should be performed with specific leptospira serovars and the end point titre will be determined.

Lines 426 to 432. I cannot fully agree with the statement that the domination of the Icterohaemorriaghae and the Pomona serovar are observed all over the world. The author cites Brazilian papers that do not reflect the global situation in the world. The Pomona and Sejroe serovars are still dominant in Europe. In addition, the presence of serovar Icterohaemorriaghae is closely related to the hygiene of production and compliance with the principles of external biosecurity rules, which are neglected in some regions of the world.

These minor comments do not affect my positive review. The authors try to compare alternative diagnostic tests in the form of recombinant LigA / BCon1-5 protein based spot tests such as Dot-ELISA and LAT as rapid screening tests with the precise MAT method. Rapid LAT and IgM Dot ELISA tests have an advantage over MAT because in a relatively short time they give a test result that allows for quick targeted therapy of pigs. MAT tests, unfortunately, require experienced personnel, maintaining leptospira cultures and the time to obtain the result is at least 5 hours. The conducted research shows the direction in which laboratories should go towards the diagnosis of leptospirosis.

Reviewer 2 Report

Dear Authors,

This is a solid study with interesting results. I have only some changes to suggest:

  • Line 150: Please describe the basic info regarding culture methodology of Leprospira serovars. In which Cat 3 lab was it performed?
  • Figure 7 shoulb be split to a Figure (left) and a Table (right).
  • Line 485: Refer an example where a similar approach has been used (using other species MHC ΙΙ isotypes) and comment what limitations can there be in this approach.

Reviewer 3 Report

Behera and colleagues perform an interesting approach on field-oriented rapid immunodiagnostic of porcine leptospirosis. I have a few recommendations for improving the overall quality of the article, including adding more state of art results about Lig proteins’ role on virulence and as a diagnostic marker. Below are some major and minor concerns to be considered by the authors:

Hsieh et al. (2017) published an elegant work employing multi-domain constructs and small angle X-ray scattering (SAXS) to determine the structure of LigB, and to determine the immunoreactive potential of each region by using a series of monoclonal antibodies. I believe that including these results in the discussion, in comparison to the presented data, would further give support to the autors’s findings.

In the Introduction, I’d suggested including the recent findings that both LigA and LigB are involved in serum resistance displayed by pathogenic Leptospira and that they are necessary for acute disease, since LigA and B silencing by novel CRISPR interference resulted in asymptomatic infection in hamster model (Fernandes et al., 2021). Those findings would reinforce the rational of using Lig proteins for diagnosis.  

Including the LigA/BCon1-5 conservation among distinct Leptospiral species would again reinforce the use of this proteins and would offer an additional advantage regarding MAT test.

Regarding the collected sera, were the animals already vaccinated for leptospirosis? Did the author checked if indeed, vaccinated-animal serum can react or not with LigCon1-5?

Line 155: I believe that the 1:100 format is more usual than “1 in 100”. This occurs several other times throughout the text. I suggested standardization for the 1:100, 1:1200, etc.. format

Line 156: Please, include more information about leptospiral cultures. Is it EMJH media? In house or Difco’s? Stationary or shaking?

Line 225: The fact that hyperimmune sera against L. icterohaemorrhagiae (is it correct, by the way?) can so well react to the transferred LigCon1-5 isn’t contrary to the suggested DIVA potential of this fragment?

The secondary antibody is anti-IgM or anti-IgG?

Line 252: Which volume was used to make the immobilization?

Why didn’t the author also try to include diagnostic with anti-IgG antibodies? Depending on the kinetics of infection, IgG antibodies would be more reactive to the antigen.

Line 371. The authors mentioned that “…reached the maximum level at 8h post induction”. Is it data not shown? Maybe it would be better to include “data not shown”.

Line 378: Again, is it anti-IgM or anti-IgG?

Line 426: I believe is better to express the predominant serovar in regards to the number of reactive samples. For instance, 143 in 275.

Please, include in the text the expected molecular weight of the recombinant protein. Was it soluble or insoluble (inclusion body)?

Minor issues

Line 257. There is a @ symbol

Line 297. “A total of 384. 385…” should be 384, 385…

Texts in several pictures present the underline red line from Microsoft word, e.g., Figure 5 “epitopes” and “Ellipro”. Please, check.

Line 368: mg/L.
